# Safety and Metabolic Tolerance of Citrate Anticoagulation in Critically Ill Polytrauma Patients with Acute Kidney Injury Requiring an Early Continuous Kidney Replacement Therapy

**DOI:** 10.3390/biomedicines11092570

**Published:** 2023-09-19

**Authors:** Filippo Mariano, Alberto Mella, Paolo Randone, Fulvio Agostini, Daniela Bergamo, Maurizio Berardino, Luigi Biancone

**Affiliations:** 1Department of Medical Sciences, University of Turin, 10126 Turin, Italy; alberto.mella@unito.it (A.M.); dbergamo@cittadellasalute.to.it (D.B.); luigi.biancone@unito.it (L.B.); 2Nephrology, Dialysis and Transplantation U, University Hospital City of Science and Health, CTO Hospital, 10126 Turin, Italy; paolo.randone@unito.it; 3Anesthesia and Intensive Care 3, University Hospital City of Science and Health, CTO Hospital, 10126 Turin, Italy; fagostini@cittadellasalute.to.it (F.A.); mberardino@cittadellasalute.to.it (M.B.)

**Keywords:** polytrauma patients, early AKI, continuous kidney replacement therapy, regional citrate anticoagulation, rhabdomyolysis, ultrafiltration rate, circuit citratemia

## Abstract

For severe polytrauma patients with an early AKI requiring renal replacement therapy, anticoagulation remains a great challenge. Due to a high bleeding risk, hemodynamic instability, and increased lactate levels, continuous modality (CKRT) and citrate anticoagulation seem to be the most appropriate. However, their safety with regard to the potential risk of impaired citrate metabolism is not documented. A retrospective study of 60 severe polytrauma patients admitted to the emergency department between January 2000 and December 2021 was conducted; the patients requiring CKRT during the first 72 h were treated with citrate (n. 46, group Citrate) or with heparin (n. 14, group Heparin). Out of 60 patients, 31 survived (51.7%). According to logistic regression analysis, age and SOFA score were significant predictors of mortality. The incidence of rhabdomyolysis was more common in the survivors (77.4 vs. 51.7%), and Kaplan–Meyer analysis showed a better trend towards survival at 90 days for the group Citrate than the group Heparin (*p* 0.0956). In the group Citrate, hemorrhagic episodes were significantly less common (0.045 vs. 0.273 episodes/day, *p* < 0.001); the effective duration (h/day) of CKRT was longer; and the effective net ultrafiltration rate (mL/kg/h) and blood flow rate were lower. For severe polytrauma patients, early, soft CKRT with citrate anticoagulation at a low blood flow rate and circuit citratemia showed a better safety and hemodynamic stability, suggesting that citrate should be the first choice anticoagulant in this subset of patients.

## 1. Introduction

An early acute kidney injury (AKI) is a common and severe complication in polytrauma patients, who are burdened by increased mortality [1]. The incidence varies from 6 to 50%, with an onset mean time of three days from the trauma [2,3]. AKI often arises in patients with extensive rhabdomyolysis and massive release of myoglobin [4], in combination with uncontrolled bleeding, coagulopathy, inflammation, abdominal hypertension, and other external injury factors, such as radiographic contrast agents. Fortunately, in the total trauma population, the proportion of the most severely injured who require kidney replacement therapy (KRT) because of oligo-anuria, fluid overload, severe acidosis, or hyperkalemia is estimated to be about 1–2% [1,2,3,5,6].

In the first days of admission, the application of KRT remains a significant challenge in polytrauma patients. Polytrauma patients are usually hemodynamically unstable, shocked, unable to tolerate high solute and fluid removal rates, and suffering from alterations in platelet count and function, consumption of coagulation factors, and active bleeding and often need massive transfusions. Therefore, the most appropriate renal replacement therapy should be the continuous replacement therapy modality (CKRT), avoiding heparin due to the unbearable increased hemorrhagic risk [7]. To maintain the long-term extracorporeal circuit patency and continuous modality, regional citrate anticoagulation could be either an attractive alternative to heparin [8,9,10] or a hazard, due to the risk of citrate accumulation in patients with a marked deficit of peripheral perfusion and increasing lactate levels [11,12].

To the best of our knowledge, the data available on trauma patients treated by CKRT in the first days after trauma are limited: in the last ten years, a few papers have described RRT experiences (five with CKRT modality, one with prolonged intermittent renal replacement therapy, and one with mixed intermittent hemodialysis and CKRT [13,14,15,16,17,18,19]), and no study has reported the dialysis methodology, the detailed type of applied anticoagulation, and the safety of the treatment.

In this study, we reported the experience of 60 severely injured polytrauma patients with AKI undergoing CKRT during the first 72 h after admission. We analyzed the survival rate and risk factors for mortality in patients treated with heparin or citrate, focusing on the safety and metabolic tolerance of citrate anticoagulation during the first 12 days of CKRT.

## 2. Materials and Methods

### 2.1. Study Population

Between January 2000 and December 2021, 1617 adult (≥18 y) patients were admitted to the emergency department of CTO Hospital, Turin, with polytrauma [20]. Among these, 62 (3.83%) required KRT during the first 72 h after admission. Polytrauma patients with associated burns were excluded from the analysis, as were 2 patients treated with early CKRT because of chronic regular dialysis (see Figure 1). As the standard anticoagulation protocol until 2006, polytrauma patients needing early CKRT were treated with unfractionated heparin. In November 2006, an off-label regional citrate anticoagulation protocol became available [8], and since then, almost all admitted patients have been treated with the citrate anticoagulation protocol. 

Out of 60 patients treated from 2000 to 2021, 46 received citrate (the group Citrate) and 14 received heparin (the group Heparin).

The study was conducted according to the Helsinki Declaration and approved by the ethics committee of our hospital (dossier n. CS2/908 on 6 August 2018). The retrospective review of the medical notes and the collected data were treated according to the regulatory rules of the ethics committee.

### 2.2. Data Collection and Management of Trauma

We collected baseline demographic characteristics, biochemical data, and dialytic parameters over the CKRT period. A multidisciplinary team managed the systemic diagnosis and treatment of the trauma patients based on the guidelines in place at that time [21,22,23]. The prevention of AKI associated with rhabdomyolysis involves recognizing and treating the underlying cause (relief of compartment syndrome), urinary alkalinization, and the maintenance of urine output higher than 2 mL/kg with the use of loop diuretics and isotonic bicarbonate solution infusion [24]. Mortality was calculated as in-hospital mortality at 90 days after admission.

Rhabdomyolysis was defined as the peak plasma myoglobin concentration exceeding 10,000 ng/L [25]. However, when we have only the determination of the CK level, specific management protocols are often initiated when the creatine kinase level is greater than 5000 U/L. Hemorrhagic episode (number of episodes/patient/day) during the first 6 days of CKRT was defined as bleeding resulting in death; retroperitoneal, intracranial, or intraocular hemorrhage; or a transfusion requirement of more than 2 U of red cells [26].

### 2.3. Management and Technique of CKRT

CKRT was started when the patients with AKI were not responsive to maximal diuretic therapy and conservative management. The typical indications for initiation consisted of fluid overload and oliguria, severe hyperkalemia, severe acidosis, or a combination of more criteria [25]. CKRT was performed in a CVVHD/CVVHDF modality with heparin or regional citrate anticoagulation protocol (see below) using a dedicated machine (Prisma, Gambro, Lund, Sweden) or Multifiltrate (Fresenius Medical Care, Bad Homburg, Germany). CKRT was performed according to the manufacturer’s instructions with high permeability biocompatible synthetic dialysis membranes (polyacrylonitrile or polysulfone dialyzers). For the patients with documented rhabdomyolysis requiring CKRT, from 2011 our protocol envisaged the use of a medium cut-off polysulfone filter EMiC2 (Fresenius Medical Care) to increase the myoglobin loss in the effluent [27]. Blood flow and effluent rates were set according to the target of dialysis adequacy, accomplishing the dialysis target of 20–25 mL/kg/h [25]. Moreover, as these rates were occasionally limited by the efficiency and patency of the vascular access, a dedicated nurse recorded the effective flow rates. The vascular access was provided by a temporary double-lumen venous catheter inserted into the jugular or femoral vein. CKRT was discontinued for renal recovery defined as independence from CKRT or upon the patient’s death.

All the patients with both heparin and citrate were intentionally treated in a continuous modality to provide appropriate hemodynamic stability and dialysis tolerance. Dialysis tolerance was defined as a stable MAP at the desired ultrafiltration rate without the need for an increased catecholamine dosage.

### 2.4. Anticoagulation in Group Heparin

In the group Heparin, anticoagulation was achieved with unfractionated heparin (Liquemin, Roche SpA, Milan, Italy) at a low dosage, administered prefilter to an initial bolus of 1250 U followed by 250 U/h. Subsequent adjustments were made accordingly to avoid premature circuit clotting. In the presence of hemorrhagic complications, the patients of the group Heparin were shifted towards a prolonged modality without heparin and saline flushing, taking into account dialysis tolerability (net ultrafiltration rate, inotropic support, and hemodynamic parameter) and adequacy (control of uremia, acid–base status, and electrolyte alterations). Dialysis tolerability was defined as a stable MAP at the desired ultrafiltration rate without the need for an increased catecholamine dosage. A standard sterile bicarbonate-containing solution was used as a dialysate/infusion (Prismasol 32, Hospal, Mirandola, MO, Italy).

### 2.5. Anticoagulation in Group Citrate

In the group Citrate from 2007 to 2013, the citrate anticoagulation strategy was systematically applied by a previously described in-home, off-label protocol [8,9,28,29]. Briefly, the citrate source was an ACD-A solution (Fresenius Kabi Italia, Isola della Scala (VR), Italy) dispersed and infused in a predilution mode, prepared by nurses immediately before the CKRT sessions. A standard sterile bicarbonate-containing solution without calcium was used as the dialysate (Ci-Ca, Fresenius Medical Care; composition (mmol/l): Na 133, Cl− 116.5, bicarbonate 20, Mg++ 0.75, K+ 2, glucose 5.5), and when appropriate, a solution with calcium was infused post-dilution (Prismasol 32, Hospal; composition (mmol/L): Na+, 140; K+ 2.0; Ca++ 2; Mg++ 0.75; Cl−: 108; bicarbonate 32; acetate 4; glucose 5.5). To replace the calcium loss by filter a commercial 10% calcium chloride solution was infused in a separate line at the end of the venous circuit by the monitor heparin pump [9,28]. Since 2013, regional citrate anticoagulation has been carried out through the monitor Multifiltrate (Fresenius Medical Care), following the manufacturer’s instructions. In the group Citrate the target circuit citratemia varied from 2 to 6 mmol/L. The adequacy of the circuit citrate anticoagulation was determined by the measurement of the post-filter iCa++ levels (GEM 3000, Instrumentation Laboratory, Milan, Italy), with a target level of <0.3 mmol/L daily, or more often when necessary.

When the patients of the group Citrate were at risk of citrate accumulation (lactate levels > 6 mmol/L and/or with an increasing trend) [11,12], they started CKRT without an anticoagulant or with heparin at 125 U/h, and within 12 h, they shifted to citrate anticoagulation at a circuit citratemia of 2 mmol/L.

### 2.6. Statistical Analysis 

Categorical and continuous variables are expressed as percentages and frequencies and as medians with interquartile ranges, respectively. The normality of the distribution was analyzed using the Kolmogorov–Smirnov test. The chi-square or Fisher’s exact test was used to compare the categorical variables. The Student’s *t*-test or the Mann–Whitney U test was used for the continuous variables with or without normal distribution. Multivariable analysis was performed using logistic regression for the total number of non-surviving patients to determine the effect on the in-hospital mortality rate of the following variables of interest: age, SOFA score at the start of CKRT, norepinephrine requirement, lactate, and bicarbonate level. The Kaplan–Meier estimate of survival was constructed to compare 90-day survival between the patients of the group Citrate and the group Heparin. Cox’s F-test was used to test the difference in survival rates.

Statistical computing and graphics were performed by Statistica v.10.1 (Statsoft, Tulsa, OK, USA). A value of *p* < 0.05 was considered statistically significant.

## 3. Results

### 3.1. Baseline Characteristics of Survivors and Non-Survivors

Among the 60 polytrauma patients treated with early CKRT for AKI, the leading cause of injury was car accidents (31.7%), followed by falls (28.3%) and motorbike accidents (20.0%) (Figure 1). 

Of the 60 polytrauma patients, 31 survived and 29 died (a survival rate of 51.7%). 

Table 1 shows the baseline characteristics of the survivors and non-survivors. At the beginning of CKRT, all the patients were on mechanical ventilation and under inotropic therapy.

The incidence of rhabdomyolysis was detected in 65% of the patients and was significantly more common in the survivors. On the other hand, the non-survivors were significantly older, had a higher SOFA score, a norepinephrine requirement, and a significantly reduced urine output. In addition, they presented more elevated lactate and lower bicarbonate blood levels.

According to the logistic regression model, which included age, SOFA score, norepinephrine, lactates, and bicarbonates, age and SOFA score were shown to be significant predictors of mortality (Table 2).

### 3.2. Baseline Characteristics and Outcome of Patients According to Citrate and Heparin Anticoagulation

By comparing the characteristics of the 60 treated patients according to anticoagulation with citrate or heparin, no difference was found for all the previously considered variables, including age, rhabdomyolysis incidence, urine output, the median number of CKRT days, SOFA score, norepinephrine requirement, and blood lactate and bicarbonate levels (Table 3).

Survival analysis at 90 days by Kaplan–Meyer curves showed no significant difference even if there was a trend towards reduced mortality for patients of the group Citrate (*p* 0.09557, Cox’s F-test, Figure 2).

### 3.3. Dialytic and Clinical Data during the First 12 Days of CKRT

The median values of the daily CKRT hours of the group Citrate were significantly longer than those of the group Heparin (Table 4). While the net ultrafiltration rate that normalized over the 24 h was about 30 mL/kg/day and was similar in the two groups, as was the delivered dialysis dose, the effective net ultrafiltration rate expressed as mL/kg/h was significantly lower in the group Citrate (Table 4). Likewise, these dialysis exchanges were performed at a blood flow rate (mL/min) that was significantly lower in the group Citrate (Table 4). In addition, the hemorrhagic episodes during CKRT were significantly less common in the group Citrate than in the group Heparin (0.045 vs. 0.273 episode/day/patient, *p* < 0.002).

Table 5 summarizes the advantages and disadvantages of the use of citrate and heparin anticoagulation during the first 12 days of CKRT.

Figure 3, Figure 4 and Figure 5 detailed the trend of different parameters in the first 12 days of treatment for the groups Citrate and Heparin.

While the trends of the MAP and the PO_2_/FiO_2_ ratio were superimposable in the two groups (Figure 3), the blood lactate and norepinephrine requirement constantly improved over the days only in the group Citrate (Figure 3).

The daily normalized net UF rate (mL/kg/day) was higher in the group Heparin. However, due to the significantly reduced daily duration of CKRT the effective hourly net fluid removal was lower in the group Citrate; it showed a steady value over all the days and was provided at a lower blood flow rate (Figure 4). Figure 5 shows the trend of the pH and bicarbonates and the observed circuit citratemia and administered heparin amount.

## 4. Discussion

The present study showed that citrate anticoagulation is safer than heparin for polytrauma patients requiring an application of CKRT within 72 h after admission, suggesting that this protocol should be the first-choice treatment for these critically ill patients with a risk of bleeding.

Despite the prevention measures, such as early volume expansion and urinary alkalinization [30], early AKI is a common complication in polytrauma patients [2,3,4,5]. Many factors are implicated, including hemorrhagic hypovolemia, a high volume of blood transfusion as the surrogate marker of blood loss, exposure to the products of hemolysis, iodine contrast nephropathy, the early onset of sepsis, abdominal compartment syndrome, and rhabdomyolysis [5,31]. Notably, in our series of patients the incidence of rhabdomyolysis was as high as 65% of the patients; the incidence was higher in the survivors but not significantly different in the non-survivors. Therefore, it is conceivable that rhabdomyolysis could have been a real risk factor for AKI onset, but it did not affect mortality. Indeed, in these CKRT patients mortality was significantly associated with the severity of organ failure (SOFA score) and age, both of which are well-known risk factors of death in AKI and in the general polytrauma population [1,2,3,4,5].

Early KRT treatment in critically ill polytrauma patients is a great challenge. Several modalities of KRT have been proposed, from continuous (CKRT) or sustained low-efficiency dialysis to short-duration treatment, such as intermittent hemodialysis. In all of these studies, no details were reported; in most, the anticoagulation modality was unspecified, and when citrate was cited, the experience was not documented [13,14,15,16,17,18,19].

After an initial experience with the heparin anticoagulation at a low dosage, in January 2007 our group started to treat these patients with citrate anticoagulation. Citrate anticoagulation was considered for several reasons. Firstly, for hemodynamic instability it was mandatory to treat the patients in a continuous modality, often for several days with a slow and constant ultrafiltration rate. Secondly, the anticoagulation response with heparin was quite unpredictable due to the coagulation disorders always occurring, such as low platelet count, reduced hematocrit, alterations of INR and the PTT ratio, and variable antithrombin III activity for consumption or replacement infusion. For all these factors, with a continuous modality the unpredictability of the anticoagulation response to heparin significantly increased. Thirdly, to obtain appropriate circuit patency the patients experienced a requirement of about 300–400 U/h of heparin, and this regimen exposed them to an unacceptably high incidence of hemorrhagic episodes. As a result, we experienced both a high number of hemorrhagic episodes and premature circuit clotting. In effect, in agreement with previous reports [8,26,27], we observed that the safety of the treatment was significantly increased with citrate, as documented by a significant reduction in hemorrhagic complications and an increased daily hour of effective dialysis.

By choosing the citrate anticoagulation in polytrauma patients, citrate metabolism was the first concern. The citrate returning to the patient was normally metabolized in the Krebs cycle in the liver, skeletal muscle, and kidney, yielding three mmol of NaHCO_3_ from one mmol of citrate. Shocked polytrauma patients showed increased lactate levels, metabolic acidosis, and sometimes liver failure with high transaminases. The metabolic impairment of citrate metabolism could lead to its accumulation, worsening the lactic acidosis and anion gap [32]. To circumvent this issue, in the first days of CKRT the amount of delivered citrate was minimized with a low blood flow rate of 80–90 mL/min and a circuit citratemia not exceeding 3.0–3.5 mmol/L (Figure 4 and Figure 5). The minimal amount of citrate infused, associated with its sustained loss in the effluent, warranted an estimated entering citrate load to the patient of no higher than 8–10 mmol/h [28]. As documented by the trend of acid–base status, lactate, MAP, and a norepinephrine requirement for up to 12 days (Figure 4 and Figure 5), this protocol of citrate anticoagulation was associated with good metabolic tolerance and constant improvement in hemodynamics.

The overall mortality of our series of 60 polytrauma patients requiring early CKRT was 48.3%. When considering the groups Citrate and Heparin separately, the mortality rates were 43.5 and 57.1%, respectively, and the 90-day K–M curve showed a better survival trend for citrate (*p* 0.0957, see Figure 2). Notably, the baseline characteristics of both groups were superimposable, and most deaths were seen in the first 12 days after the beginning of CKRT. There could be many factors contributing to the improved survival with citrate; however, no factor is easily isolable, nor can it be demonstrated to be crucial to survival in such critically ill and complex patients. Patients with heparin anticoagulation were admitted in a different period of time, and the comparison between heparin and citrate was equivalent to comparing patients between two time periods; in the years between these periods, changes in intensive care standards occurred in such areas as ventilation, indications, and management of dialysis. However, the analysis of the first 12 days of CKRT demonstrated that in the clinical practice the continuous treatment was “more continuous” for the group Citrate than for the group Heparin, which suffered from more frequent interdialytic pauses due to hemorrhagic episodes and circuit clotting. This could lead to some potential beneficial effects for the group Citrate. Firstly, even if the absolute amount of daily normalized net fluid removal was the same for both groups, the effective net fluid removal (mL/kg/h) was significantly lower in the group Citrate, as the patients of the group Heparin were treated for a shorter daily time. While fluid and blood product administration plays a crucial role in the early hours after injury to obtain hemodynamic stabilization, in the next phase deresuscitation (defined as the active fluid negative balance using ultrafiltration care in patients with fluid accumulation) deserves the same great importance [3,33]. It is conceivable that a steady, soft net fluid removal could contribute to better hemodynamic stability, as documented by an increased requirement of norepinephrine and higher lactate levels in the group Heparin since the 3rd–4th day of CKRT. Secondly, the heparin amount used was minimal, and a sustained blood flow rate was chosen to increase the patency of the extracorporeal circuit [34,35]. It has been documented that a higher intensity of CKRP could lead to several drawbacks, such as electrolyte imbalances, rapid shifts in serum osmolality among the body compartments, the loss of nutrients or thermal energy, and underdosing of antimicrobial agents [36]. Conversely, in the group Citrate, the blood flow rate was low to minimize the load of citrate, and this soft dialysis could play a role in ensuring the observed association with better systemic hemodynamic stability. Thirdly, it is well known that polytrauma patients present a state of severe endothelial injury, coagulation derangements, and systemic inflammation [37]. In this clinical context, some evidence supported the idea that citrate, in comparison with heparin, could improve the biocompatibility of the extracorporeal circuit, limiting the interaction between the filter membrane and the blood components [38].

We are aware that our study had several limitations. It was a single-center report and a retrospective, with all the related drawbacks. The patients were admitted over a long period of time; during this time, changes in intensive care standards had occurred over the years, in addition to changes in the management of dialysis. However, for the present “niche work” on the patients treated with citrate anticoagulation within the first 72 h, to the best of our knowledge the number of recruited subjects was quite large. The better favorable trend in dialytic parameters and survival with citrate could have a great clinical potential value because of the homogeneity of the population; the patients of this population were treated for a specific complication, at a specific time after hospital admission, with a clearly defined protocol of dialytic technique.

Even if further studies and more data are needed to confirm these results, this should encourage efforts toward more precision for continuous replacement therapy and solute control [39].

## 5. Conclusions

The present report detailed the feasibility of citrate anticoagulation in a large cohort of polytrauma patients needing early CKRT. It documented the citrate metabolic tolerance and a better safety and hemodynamic stability for citrate, suggesting that it should be the first-choice anticoagulant in this subset of patients.

## Figures and Tables

**Figure 1 biomedicines-11-02570-f001:**
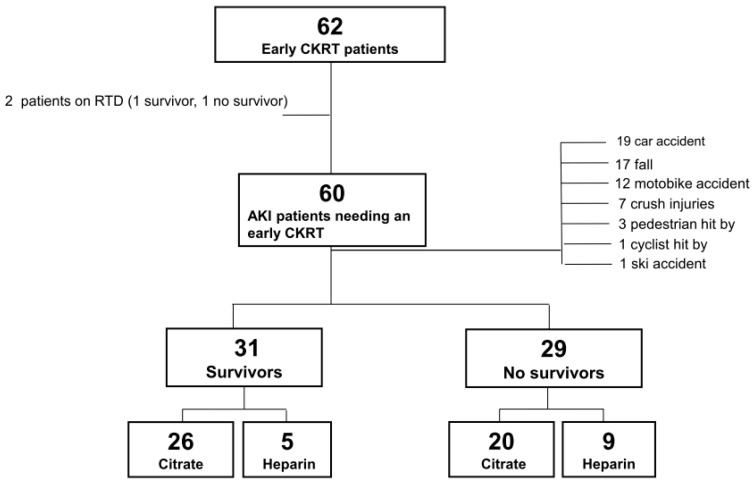
Flow chart of trauma patients admitted to emergency department and undergoing early continuous renal replacement therapy <72 h after admission.

**Figure 2 biomedicines-11-02570-f002:**
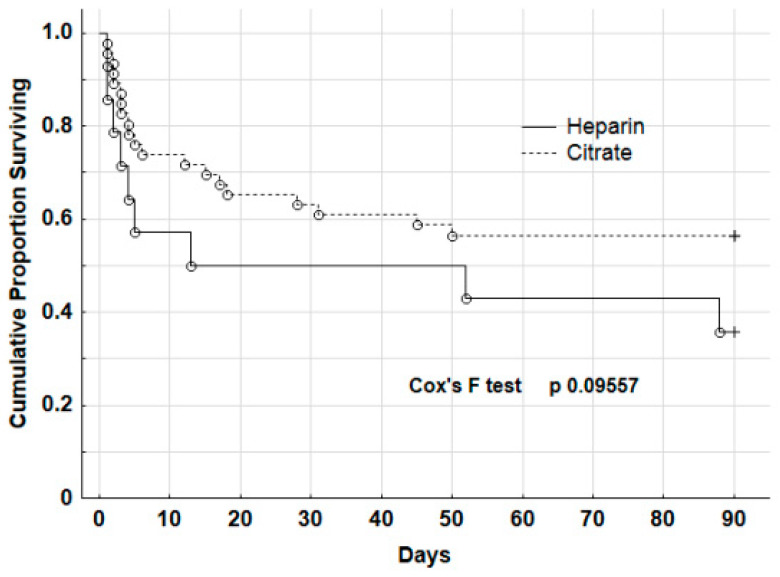
Survival analysis by Kaplan–Meyer curves for the group Citrate (n. 46 patients) and the group Heparin (n. 14 patients).

**Figure 3 biomedicines-11-02570-f003:**
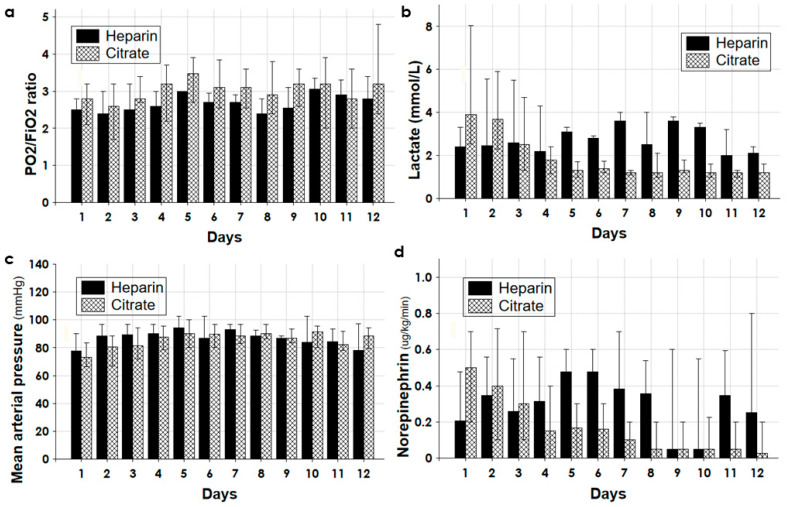
(**a**) Ratio pO_2_/FiO_2_, (**b**) systemic blood lactate (mmol/L), (**c**) mean arterial pressure (MAP) mmHg, and (**d**) norepinephrine requirement (μg/kg/min) during the first 12 days of treatment for the groups Citrate and Heparin.

**Figure 4 biomedicines-11-02570-f004:**
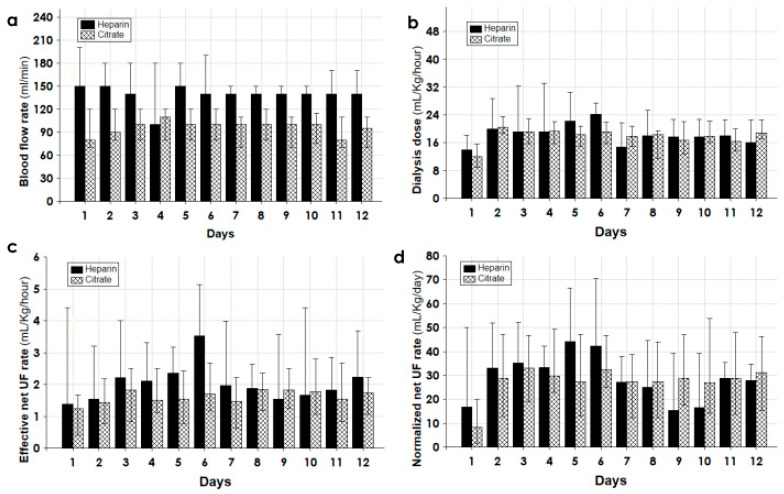
(**a**) Blood flow rate (mL/min), (**b**) dialysis dose (mL/kg/h), (**c**) effective net ultrafiltration rate (mL/kg/h), and (**d**) normalized net ultrafiltration rate (mL/kg/day) during the first 12 days of treatment for the groups Citrate and Heparin.

**Figure 5 biomedicines-11-02570-f005:**
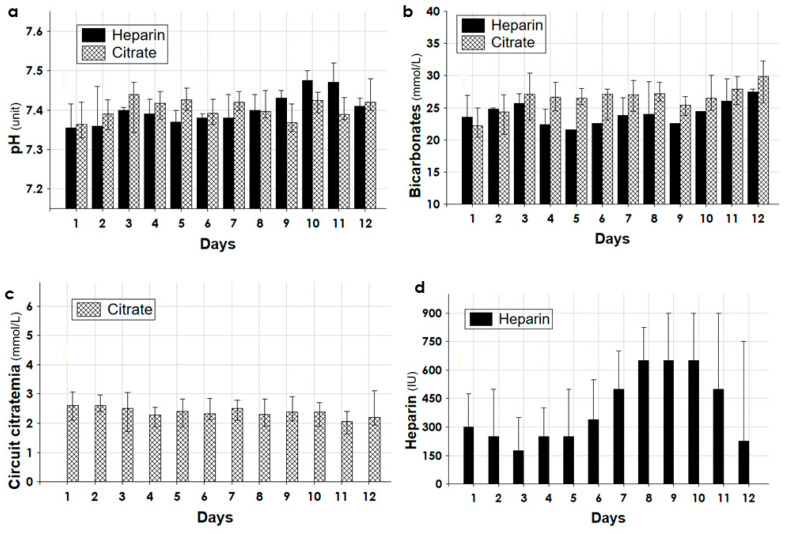
(**a**) pH (units), (**b**) bicarbonates (mmol/L), (**c**) circuit citratemia (mmol/L), and (**d**) heparin dosage (units/h) during the first 12 days of treatment for the groups Citrate and Heparin.

**Table 1 biomedicines-11-02570-t001:** Baseline characteristics of the 60 polytrauma patients with AKI requiring CKRT within 72 h according to survival.

	All	Survivors	Non-Survivors	*p*
Patients (%, n)	60	51.7%, 31	48.3%, 29	-
Age (years)	54 (38.5–72)	45.0 (31–68)	68.0 (49–78)	0.002
Gender ratio (male/female)	55/5	28/3	27/2	0.699
Mechanical ventilation (%, n)	100%, 60	100%, 31	100%, 29	-
Rhabdomyolysis (%, n)	65.0%, 39	77.4%, 24	51.7%, 15	0.037
Urine output (mL/day)	180 (0–935)	390 (0–2400)	0.0 (0–250)	0.003
CKRT days	768	606	162	-
Median CKRT time (days)	10 (3–18)	11 (5–22)	3 (2–13)	0.035
CKRT interval (days after admission)	2 (2–3)	2 (2–3)	2 (2–3)	0.810
Citrate anticoagulation (%, n)	76.7%, 46	81.2%, 26	71.2%,20	0.369
Exitus interval (days after admission)	-	-	4.0 (2–17)	-
on 1st day of CKRT ^a^				
SOFA score	13 (11–15)	12 (11–14)	15 (13–16)	0.006
Cardiovascular	4 (4–4)	4 (1.5–4)	4 (4–4)	0.015
Respiratory	2 (1–3)	2 (1–2)	2 (1.5–3)	0.011
Hematologic	2 (1–2)	2 (1–2)	2 (1–2)	0.431
Liver	2 (1–2)	1 (1–2)	2 (2–2)	0.060
Norepinephrine (ug/kg/min)	0.4 (0.2–0.7)	0.35 (0.1–0.5)	0.5 (0.3–1.0)	0.025
Lactate (mmol/L)	3.4 (2.1–8.0)	2.8 (1.9–4.6)	4.6 (2.4–10.1)	0.014
Bicarbonates (mmol/L)	22.4 (19.4–25.0)	23.0 (21.6–25.4)	20.8 (16.9–25.0)	0.005

^a^ Data were taken at the beginning of CKRT. Data are given as median (the 25th and 75th percentiles) or percentage and frequency. Student’s t or Pearson chi-square test was performed when appropriate.

**Table 2 biomedicines-11-02570-t002:** Logistic regression analysis for the predictors of mortality.

	Odds Ratio	Confidence Intervals		*p*-Value
		−95% CL	+95% CL	
SOFA score (on the 1st day)	1.472	1.076	2.014	0.013
Age	1.049	1.008	1.092	0.016
Norepinephrine	1.459	0.284	7.494	0.643
Lactates	1.020	0.831	1.252	0.843
Bicarbonates	0.886	0.746	1.052	0.159

**Table 3 biomedicines-11-02570-t003:** Baseline characteristics of 60 polytrauma AKI patients according to anticoagulation with citrate (n. 46 patients) and heparin (n. 14 patients).

	Group Citrate			Group Heparin			
	All	Dead	Alive	All	Dead	Alive	*p*
Patients (%, n)	46	43.5%, 20	56.5%, 26	14	57.1%, 9	42.9%, 5	0.121
Age (years)	54.5 (41–72)	66 (54–79)	46 (38–68)	50 (26–71)	71 (35–73)	26 (25–54)	0.371
Sex ratio (male/female)	42/4	19/1	23/3	13/1	8/1	5/0	0.854
Mechanical ventilation (%, n)	100%, 46	100%, 20	100%, 26	100%, 14	100%, 9	100%, 5	-
Rhabdomyolysis (%, n)	63.1%, 29	50.0%, 10	73.1%, 19	71.4%, 10	55.5%, 5	80.0%, 4	0.565
Median CKRT time (days)	11.0 (4–21)	4.5 (3–18)	12.5 (5–22)	7.0 (1–13)	3.0 (1–12)	10 (8–13)	0.781
CKRT interval (days after admission)	2.0 (2–3)	2.0 (2–3)	2.0 (2–3)	3.0 (2–3)	2.0 (1–3)	3.0 (3–3)	0.745
Exitus interval (days post-CRRT)	-	4.5 (2.5–17.5)	-	-	4.0 (2.0–13)	-	0.466
Urine output (mL/day)	158 (0–890)	375 (0–2400)	0 (0–230)	290 (0–1600)	150 (0–400)	1600 (860–2400)	0.551
on 1st day of CKRT ^a^							
SOFA score	13.0 (12–15)	15.0 (13–16)	12.0 (10–14)	13.5 (11–16)	15.0 (13–16)	12.0 (10–14)	0.368
Cardiovascular	4 (4–4)	4 (4–4)	4 (3–4)	4 (4–4)	4 (4–4)	4 (0–4)	0.542
Respiratory	2 (1–3)	2 (1–3)	1 (1–2)	2 (2–3)	2 (2–3)	2 (2–3)	0.059
Hematologic	2 (1–2)	2 (1–2.5)	2 (1–2)	2 (1–2)	2 (1–2)	2 (2–2)	0.676
Liver	2 (1–2)	2 (1–2)	1.0 (1–2)	2 (1–2)	2 (1–2)	2 (1–2)	0.683
Norepinephrine (μg/kg/min) ^a^	0.43 (0.2–0.7)	0.5 (0.3–1.0)	0.38 (0.1–0.6)	0.35 (0.1–0.7)	0.47(0.2–0.7)	0.18 (0.1–0.5)	0.385
Lactate (mmol/L) ^a^	3.9 (2.1–8.0)	5.1 (2.6–10.1)	3.0 (1.9–4.6)	2.5 (2.1–4.8)	2.2 (2.2–2.6)	3.1 (2.4–8.1)	0.966
Bicarbonates (mmol/L) ^a^	22.1 (18–25)	20.5 (16–25)	22.6 (22–25)	23.0 (21–25)	21.6 (17–23)	25.0 (24–25)	0.685

^a^ Data were taken at the beginning of CKRT. Data are given as median (quartile 1–quartile 3) or percentage and frequency. The chi-square test or Fisher’s exact test, and Student’s t test or the Mann–Whitney U test were performed between citrate (all cases) and heparin groups (all cases) when appropriate.

**Table 4 biomedicines-11-02570-t004:** Flow rates and dialytic data according to citrate and heparin anticoagulation during the first 12 days of CKRT.

	All	Group Citrate	Group Heparin	*p*
CKRT days	418	306	112	-
Effective time of CKRT (h/day)	20.0 (11–23)	20.0 (12–23)	16.0 (7–22)	0.002
Filter lifespan (h)	36.0 (24–48)	38.5 (24–49)	33.0 (24–41)	0.250
Blood flow rate (mL/min)	100 (80–120)	100 (80–120)	140 (100–180)	0.002
Delivered dialysis dose (mL/kg/h)	18.0 (13.2–22.2)	18.1 (13.3–21.2)	17.8 (12.8–24.0)	0.075
Normalized net fluid removal (mL/kg/day)	28.0 (11.1–46.0)	27.5 (11.8–45.7)	33.1 (9.8–49.7)	0.308
Effective net fluid removal (mL/kg/h)	1.61 (0.83–2.50)	1.53 (0.76–2.35)	2.03 (1.40–3.39)	0.002
Circuit citratemia (mmol/L)	-	3.5 (2.9–4.0)	-	-
Unfractionated heparin (units/h)	-	-	350 (150–650)	-
Hemorrhagic episodes (1/days, n)	0.103, 43	0.045, 14	0.273, 29	0.002

Data are given as median (the 25th and 75th percentiles) or percentage and frequency. The chi-square test or Fisher’s exact test, and Student’s t test or the Mann–Whitney U test were performed between the groups Citrate (all cases) and Heparin (all cases) when appropriate.

**Table 5 biomedicines-11-02570-t005:** Advantages and disadvantages of the use of citrate and heparin anticoagulation during the first 12 days of CKRT.

Parameter	Citrate	Heparin
Daily effective time	high, dialysis more continuous	decreased, with more down time
Prescribed fluid removal (mL/kg/daily)	potentially high	potentially high
Effective net fluid removal (mL/kg/h)	low, and constant	variable, and higher
Blood flow rate (mL/min)	low, enough at 80–100 mL/min	higher, at 120–140 to increase circuit patency
Metabolic tolerance	good, with reduced blood flow rate (60–80 mL/min) and low circuit citratemia (2 mmol/L)	good
Hemodynamic stability	good, constant over time	variable, depending on prescribed UF and blood flow rate
Circuit anticoagulation efficacy	high, predictable	less predictable (depending on platelet count, ATIII), poor with no heparin/saline flushes
Hemorrhagic episodes	low incidence	increased incidence
Nursing workload	Low	Increased

## Data Availability

The datasets used and/or analyzed during the current study are available from the corresponding author on reasonable request.

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
