# Peer review of "Safety and Metabolic Tolerance of Citrate Anticoagulation in Critically Ill Polytrauma Patients with Acute Kidney Injury Requiring an Early Continuous Kidney Replacement Therapy"

_biomedicines, 2023, doi:10.3390/biomedicines11092570_

Round 1
Reviewer 1 Report
The article "Safety and metabolic tolerance of citrate anticoagulation in critically ill polytrauma patients with Acute Kidney Injury requiring an early Continuous Kidney Replacement Therapy ", submitted for review, concerns the anticoagulant treatment of polytrauma patients with an early AKI requiring renal replacement therapy. The manuscript is clearly written and presented in a well-structured manner. The data are interpreted appropriately and consistently throughout the manuscript. The conclusions are consistent with the evidence and arguments presented. The topic taken up by the authors seems important and timely, but the manuscript needs minor improvement.
1. The article should be prepared according to the "Instruction for Authors" (including References).
2. Please provide the trade names of the preparations used (heparin, citrate) and producers.
3. Please improve the quality of Figures 3-5.
4. I suggest preparing a table with the advantages and disadvantages of the use of heparin and citrate derived from the authors' studies. This form of presentation would make an interesting summary.
Author Response
Dear Editor
Thank you for giving us the opportunity to revise our manuscript. We appreciate the time and effort that you and the reviewers dedicated to providing feedback on our manuscript and are grateful for the insightful comments on and valuable improvements to our paper.
We have tried our best to revise the manuscript according to the comments. The detailed point-by-point responses are listed below
Looking forward to hearing from you, thank you again very much for your kind consideration
Sincerely yours
Filippo Mariano
Reviewer #1
The article "Safety and metabolic tolerance of citrate anticoagulation in critically ill polytrauma patients with Acute Kidney Injury requiring an early Continuous Kidney Replacement Therapy ", submitted for review, concerns the anticoagulant treatment of polytrauma patients with an early AKI requiring renal replacement therapy.
The manuscript is clearly written and presented in a well-structured manner. The data are interpreted appropriately and consistently throughout the manuscript.
The conclusions are consistent with the evidence and arguments presented. The topic taken up by the authors seems important and timely, but the manuscript needs minor improvement.
We thank this reviewer for his/her suggestions and criticisms that have allowed us to improve the quality of the manuscript.
We provide herein a point-to-point answer to his/her queries. We provide the corrected version and a clean final copy of the paper. In our corrected version, you can find the comment for each correction. In the answers the numbers of page and line are referred to the corrected version.
- The article should be prepared according to the "Instruction for Authors" (including References).
1.R We now provided the article according to the "Instruction for Authors" including the References order of citations.
- Please provide the trade names of the preparations used (heparin, citrate) and producers.
2.R We now provided the trade names of the preparations used (heparin, citrate) and producers
- Please improve the quality of Figures 3-5.
3.R We have now improved the quality of the Figures 3-4-5 (see clean copy)
- I suggest preparing a table with the advantages and disadvantages of the use of heparin and citrate derived from the authors' studies. This form of presentation would make an interesting summary.
4.R We thank the Reviewer for the suggestion. We added the new Table 5 with the advantages and disadvantages of the use of heparin and citrate

Reviewer 2 Report
1. “Rhabdomyolysis was defined as the peak plasma myoglobin concentration exceed 10,000 ng/L, However, specific management protocols are often initiated when 339 the creatine kinase level is greater than 5000 U/L.” Should you use CK level instead of myoglobin for rhabdomyolysis
2. What is the definition of severe polytrauma?
3. “Blood flow and effluent rates were set by the target of dialysis adequacy, accomplishing the dialysis target of 20-25 ml/Kg/die” what is “ml/kg/die”? should it be “ml/kg/hr”
4. Did you monitor post-filter ionized calcium in citrate group?
5. “Categorical and continuous variables are expressed as medians with interquartile ranges” The descriptive statistics for categorical variables are missing
6. Should use “multivariable” instead of “multivariate”
7. Do you have information on filter lifespan between heparin and citrate?
8. Heparin and citrate were exclusively used in different time period. Therefore, the comparison between heparin and citrate are equivalent to comparing patients between two time periods. So we would expect the inherent practices that differ between two time periods, and it might be expected to note better or at least similar outcomes in citrate group (more recent time period)?
9. Please comment on your sample size.
fine
Author Response
Dear Editor
Thank you for giving us the opportunity to revise our manuscript. We appreciate the time and effort that you and the reviewers dedicated to providing feedback on our manuscript and are grateful for the insightful comments on and valuable improvements to our paper.
We have tried our best to revise the manuscript according to the comments. The detailed point-by-point responses are listed below
Looking forward to hearing from you, thank you again very much for your kind consideration
Sincerely yours
Filippo Mariano
Reviewer #2
We thank this reviewer for his/her suggestions and criticisms that have allowed us to improve the quality of the manuscript.
We provide herein a point-to-point answer to his/her queries We provide the corrected version and a clean final copy of the paper. In our corrected version, you can find the comment for each correction. In the answers the numbers of page and line are referred to the corrected version.
- Rhabdomyolysis was defined as the peak plasma myoglobin concentration exceed 10,000 ng/L, However, specific management protocols are often initiated when 339 the creatine kinase level is greater than 5000 U/L.” Should you use CK level instead of myoglobin for rhabdomyolysis
1.R We apologize for the incomplete sentence on page 2, subchapter 2.2 . The correct sentence indicating our protocol is: “Rhabdomyolysis was defined as the peak plasma myoglobin exceed 10,000 ng/L. However, when we have only the determination of creatine kinase (CK) level, specific management protocols are often initiated when the CK level is greater than 5000 U/L”
- What is the definition of severe polytrauma?
2.R The observation of Reviewer is right. We erase the term “severe”, and we add the new reference [20] according to the definition of Polytrauma (see page 2, subchapter 2.1 “Study population”, 2nd line)
- “Blood flow and effluent rates were set by the target of dialysis adequacy, accomplishing the dialysis target of 20-25 ml/Kg/die” what is “ml/kg/die”? should it be “ml/kg/hr”
3.R We apologize for the typing error. The changed the text in “ml/Kg/hour”
- Did you monitor post-filter ionized calcium in citrate group?
4.R Right, we monitored the post filter iCa++ according to manufacturer instruction. We now add the sentence on page 3, sub-chapter 2.5. “Adequacy of citrate anticoagulation was determined by the measurement of post filter iCa++ levels (GEM 3000, Instrumentation Laboratory, Milan, Italy) with a target level < 0.3 mmol/L daily, or more often when necessary”.
- “Categorical and continuous variables are expressed as medians with interquartile ranges” The descriptive statistics for categorical variables are missing
5.R We apologize for the incomplete sentence. We have now add the changed sentence “Categorical and continuous variables are expressed as percentages and frequencies, and medians with interquartile ranges, respectively”
- Should use “multivariable” instead of “multivariate”
6.R We have used “multivariable” instead of “multivariate”
- Do you have information on filter lifespan between heparin and citrate?
7.R In Table 4, line 3, we depicted filter lifespan (we have now added the suggested correct definition “filter lifespan”)
- Heparin and citrate were exclusively used in different time period. Therefore, the comparison between heparin and citrate are equivalent to comparing patients between two time periods. So we would expect the inherent practices that differ between two time periods, and it might be expected to note better or at least similar outcomes in citrate group (more recent time period)?
8.R.We thank the reviewer for this meaningful observation. We already wrote that this issue be a limitation of our study on page 10, last line and page 11, lines 1-1. We now integrated this issue in the Discussion on page 10 with the following sentence: “Patients with heparin-anticoagulation were admitted in a different period of time, and the comparison between heparin and citrate was equivalent to comparing patients between two time periods, in which the evolutions in intensive care standards have occurred over the years, including ventilation, and indications and management of dialysis. However, the analysis of the first 12 days of CKRT demonstrated that in the clinical practice the continuous treatment was “more continuous” for the group Citrate than for Heparin, which suffered from more frequent interdialytic pauses due to hemorrhagic episodes and circuit clotting.”
- Please comment on your sample size.
9.R We thank the reviewer for his suggestion. We commented in Discussion on page 11, lines 2-4: “for the present “niche work” on patients treated with citrate anticoagulation within the first 72 hour, to our best knowledge the number of recruited subjects was quite large.”

Round 2
Reviewer 2 Report
all of my comments have been addressed.